# Diacylglycerol Acyltransferase 3(DGAT3) Is Responsible for the Biosynthesis of Unsaturated Fatty Acids in Vegetative Organs of *Paeonia rockii*

**DOI:** 10.3390/ijms232214390

**Published:** 2022-11-19

**Authors:** Longyan Han, Yuhui Zhai, Yumeng Wang, Xiangrui Shi, Yanfeng Xu, Shuguang Gao, Man Zhang, Jianrang Luo, Qingyu Zhang

**Affiliations:** 1College of Landscape Architecture and Arts, Northwest A&F University, Yangling, Xianyang 712100, China; 2Oil Peony Engineering Technology, Research Center of National Forestry Administration, Yangling, Xianyang 712100, China; 3National Engineering Research Center for Floriculture, College of Landscape Architecture, Beijing Forestry University, Beijing 100010, China

**Keywords:** PrDGAT3, *Paeonia rockii*, TAG accumulation, fatty acid biosynthesis

## Abstract

‘Diacylglycerol acyltransferase (DGAT)’ acts as a key rate-limiting enzyme that catalyzes the final step of the *de novo* biosynthesis of triacylglycerol (TAG). The study was to characterize the function of the *DGAT3* gene in *Paeonia rockii*, which is known for its accumulation of high levels of unsaturated fatty acids (UFAs). We identified a *DGAT3* gene which encodes a soluble protein that is located within the chloroplasts of *P. rockii*. Functional complementarity experiments in yeast demonstrated that *PrDGAT3* restored TAG synthesis. Linoleic acid (LA, C18:2) and α-linolenic acid (ALA, C18:3) are essential unsaturated fatty acids that cannot be synthesized by the human body. Through the yeast lipotoxicity test, we found that the yeast cell density was largely increased by adding exogenous LA and, especially, ALA to the yeast medium. Further ectopic transient overexpression in *Nicotiana benthamiana* leaf tissue and stable overexpression in *Arabidopsis thaliana* indicated that *PrDGAT3* significantly enhanced the accumulation of the TAG and UFAs. In contrast, we observed a significant decrease in the total fatty acid content and in several major fatty acids in *PrDGAT3*-silenced tree peony leaves. Overall, *PrDGAT3* is important in catalyzing TAG synthesis, with a substrate preference for UFAs, especially LA and ALA. These results suggest that *PrDGAT3* may have practical applications in improving plant lipid nutrition and increasing oil production in plants.

## 1. Introduction

*Paeonia rockii* is a promising woody oil crop which belongs to the family Paeoniaceae [1]. In addition to high ornamental value, its oil showed antibacterial, anti-inflammatory, immunity-enhancing, and other pharmacological effects. As a “new resource food” [2], *P. rockii* seeds are rich in unsaturated fatty acids, including α-linolenic acid (ALA), linoleic acid (LA), and oleic acid (OA), which account for up to 90% [3,4]. ALA is a type of ω-3 fatty acid which is essential to health but cannot be synthesized by the human body. It has excellent functions in anti-oxidation, lowering blood lipids and preventing cardiovascular and cerebrovascular diseases. Thus, exploring the oil synthesis mechanism is important in facilitating the oil production of *P. rockii* through genetic improvement.

Triacylglycerol (TAG) is the main storage form of vegetable oils [5]. Eukaryotes synthesize TAG mainly through two pathways (Appendix A). The first pathway is the Kennedy Pathway, which relies on acyl-CoA and involves three successive acylation reactions to the glycerol main chain. The reaction begins with the formation of lyso-phosphatidic acid from glycerol-3-phosphate catalyzed by glycerol-3-phosphate acyltransferase (GPAT), which is further acylated to phosphatidic acid; then, the phosphatidic acid is converted to diacylglycerol under the action of phosphatidic acid phosphatase. Diacylglycerol acyltransferase (DGAT) acts as a key rate-limiting enzyme in plant storage lipid accumulation. In the last step of the acylation reaction, DGAT catalyzes the generation of TAG in the sn-3 position of diacylglycerol [6,7]. The second pathway is the acyl-coA independent synthesis pathway, which uses phospholipids as the lipoyl donor and diacylglycerol as the lipoyl receptor [8,9]. Phospholipids: diacylglycerol acyltransferase (PDAT) transfers the phospholipids on the sn-2 fatty acids to the acyl glycerin sn-3; the formation of TAG and lyso-phospholipids occurs mainly in yeast and vascular plants [10,11].

In general, DGAT and PDAT, the key rate-limiting enzymes in the last acylation reaction of the TAG synthesis, play a crucial role in guiding the generation of TAG from the carbon sources [12]. Current studies have shown that there are at least three different DGATs in eukaryotes: DGAT1, DGAT2, and DGAT3 [13,14,15]. DGAT1 and DGAT2 are located on the endoplasmic reticulum (ER) [16] and are integral membrane proteins. DGAT3 is a soluble protein located in the cytoplasm [17,18,19]. It is generally believed that DGAT affects seed oil content, TAG content, fatty acid composition, and seed weight during seed development [20]. Therefore, it is possible to improve the oil content and fatty acid composition of oil-bearing crops by exploring the genes encoding active protein with *DGAT* [8,21,22,23]. At present, the genes encoding functional *DGAT1* or *DGAT2* are isolated from a variety of plants, such as *Arabidopsis thaliana*, tobacco (*Nicotiana tabacum L*), peanut (*Arachis hypogaea*), rape (*Brassica napus L*), and *Sesamumindicum.* Their role in plant TAG biosynthesis has been demonstrated through overexpression and mutation downregulation techniques. For example, the *AtDGAT1* mutant *AS11* displayed reduced seed oil content by 20–40%, while the overexpression of *AtDGAT1* resulted in a 9–12% increase in seed oil content and grain weight [24]. *UrDGAT2* from *Umbelopsis ramanniana* was expressed by seed-specific promoters in soybean (*Glycine max*), generating a new soybean variety with significantly higher oil content [25]. Previous studies showed that *DGAT1*, *DGAT2*, and *DGAT3* share little sequence homology with each other, indicating that they probably have different origins in plants [19,26]. *DGAT1*, *DGAT2*, and *DGAT3* also show different expression patterns across different plant organs and tissues. The *DGAT3* gene remains as a single copy in plants, except for a few species [26]. The soluble protein DGAT3 was first reported to be purified from the cytoplasm of the developing cotyledons of peanut (*Arachis hypogaea*) [18]. In *Arabidopsis thaliana*, *AtDGAT3* can recycle LA and ALA into TAGs when the process of TAG decomposition is blocked [14]. Moreover, *AtDGAT3* has recently been shown to be a metalloprotein involved in TAG biosynthesis [27]. During cotton (*Gossypium*) seed development, the overexpression of the *GhDGAT3D* gene leads to the increased accumulation of the TAG and C18:1 content [28]. In Camellia (*Camelina sativa*), *CsDGAT3-3* also showed a strong catalytic capacity for TAG synthesis and a substrate preference for eicosenoic acid (EA) [29]. Nevertheless, the functions of DGAT1 and DGAT2 have been explored in a variety of plants, whereas the DGAT3 gene has only been characterized in few plant species.

Our study was to explore the function of the DGAT3 family members in *Paeonia rockii.* We identified and functionally characterized the *PrDGAT3* gene in *P. rockii*. Then, we performed a bioinformatics analysis of DGAT3 and revealed its expression profiles among different tissues of *P. rockii* by qRT-PCR. Meanwhile, we examined the function of *PrDGAT3* with stable transformation in *Arabidopsis thaliana*; transient transformation in *Nicotiana benthamiana* leaves; overexpression in yeast (*Saccharomyces cerevisiae*) mutant H1246; and gene silencing in *P. rockii* leaves. Finally, we showed that *PrDGAT3* is a contributor to the oil synthesis of *P. rockii* and possibly has a certain relationship with the high content of ALA. Our findings provide an insight into the role of the *DGAT3* gene and a new perspective on the improvement of lipid nutrition in plants.

## 2. Results

### 2.1. Molecular Characterization of DGAT3 from P. rockii

The length of the protein encoded by *PrDGAT3* is 390 aa. The analysis of the fatty acid index and the hydrophilicity index showed that PrDGAT3 is a water-soluble protein, lacking any transmembrane structure. The signal peptide prediction analysis reported that PrDGAT3 most likely contains a chloroplast transport peptide at the N-terminus. As shown in Figure 1A, PCAMBIA2300-*PrDGAT3*-*GFP* was the target experimental group. The subcellular localization experiment confirmed that the PrDGAT3 protein is localized in the chloroplasts. The conserved motifs of PrDGAT3 were identified with the MEME program and were compared with motifs of the DGAT3 protein from other species. Most of the motifs are highly conserved in DGAT3, including phosphorylation sites, tyrosine kinase phosphorylation conserved sites, and putative DGAT catalytic motifs (Appendix A).

The Conserved Domain Database (CDD) was employed to identify the conserved functional domains of PrDGAT3. The results showed that the PrDGAT3 protein contains the typical domains of the thioredoxin-like [2Fe-2S] ferredoxin (TRX-like Fd) family, which suggested that PrDGAT3 may be involved in the binding of the [2Fe-2S] cluster. This domain was identified as being conserved among the DGAT3 family proteins.

In order to understand the evolution of DGAT3, we constructed a phylogenetic tree based on the protein sequences of the PrDGAT3 and DGAT3 proteins of other plant species. As shown in Figure 1C, it is noteworthy that the *PrDGAT3* of *P. rockii* was closed in relation to the DGAT3 proteins from the *Vitis riparia* and *Vitis vinifera*, forming a separate cluster, suggesting that *PrDGAT3* may share a common evolutionary ancestor with other *Vitis L.* species. These sequence analysis results confirmed that the cloned *PrDGAT3* from *P. rockii* was of full-length. Based on previous functional validation in other species, we predicted that *PrDGAT3* might be involved in the soluble TAG synthesis pathway [26,27].

### 2.2. Expression Profile of PrDGAT3

In order to explore the function of *PrDGAT3* in the oil synthesis of *P. rockii*, the expression profile of *PrDGAT3* in different tissues of *Paeonia rockii* was studied by qRT-PCR, including roots, stems, leaves, petals, and seeds. The 18S-26S internal transcribed spacer (18S-26S ITS) gene of *P. rockii* was used as an internal reference to calculate the relative gene expression in each tissue (Appendix A). As shown in Figure 1B, the *PrDGAT3* gene expression was significantly different among the organs. *PrDGAT3* was highly expressed in the stems and leaves but was barely detected in the petals. These results suggested that *PrDGAT3* was highly expressed in the green tissues, which may be involved in the lipid synthesis in chloroplasts.

### 2.3. Overexpression of PrDGAT3 Functionally Recovers oil Biosynthesis in TAG-deficient S. cerevisiae H1246

The yeast mutant H1246 has four TAG biosynthesis genes knocked out, including *DGA1* (encoding DGAT), *LRO1* (encoding PDAT), and *ARE1* (encoding acyl-CoA: Sterol transferase 1 (ASAT1), and *ARE2* (encoding ASAT2) and therefore lacks the ability to synthesize neutral oil (TAG); as a result, this mutant strain is fit for verifying the function of enzymes such as DGAT or PDAT [30]. The yeast expression vector pYES2 containing *PrDGAT3* was constructed first and was then transformed into the yeast mutant strain H1246. After 48 h of culture, lipid extraction was performed. The lipid separation and Nile red fluorescence determination were performed by TLC and with a confocal laser scanning microscope. The H1246 strain and the wild-type yeast (INVSc1) transformed with an empty pYES2 vector were used as the negative and positive controls, respectively.

As shown in Figure 2A, strong and dense fluorescence signals were observed in the positive control yeast cells and also in the H1246 yeast cells overexpressing *PrDGAT3*, demonstrating that a large number of oil bodies were formed in the transgenic yeast cells expressing the target gene. However, there was only a very weak fluorescence signal in the negative control cells and almost no oil body was generated. To further confirm this, TLC observations were performed (Figure 2B), and no TAG spot was detected in the negative control H1246 yeast cells containing the empty pYES2 vector. In contrast, there was an obvious TAG spot in the H1246 yeast cells expressing *PrDGAT3* and the positive control cells. This indicated that *PrDGAT3* expression restored the ability of H1246 yeast cells to synthesize TAG. In conclusion, the expression of *PrDGAT3* successfully restored TAG synthesis and accumulation in H1246 yeast cells, which fully suggested that *PrDGAT3* has DGAT activity.

To further examine the substrate preference of *PrDGAT3* encoded enzymes, the fatty acid content and composition of the TAG synthesized in these yeast cells were examined by GC analysis by scraping the TAG spots off the TLC plate. As shown in Figure 2C, the H1246 yeast cells overexpressing *PrDGAT3* significantly increased the total fatty acid content compared with the negative control. The oil content of the transgenic H1246 yeast cells was about one-third of that in the positive control yeast cells.

### 2.4. PrDGAT3 Manifests High Substrate Preference for LA and ALA Revealed by Yeast Lipid Toxicity Test

The Nile red observations showed that although the TAG content of the H1246 yeast cells expressing *PrDGAT3* was significantly increased, it was still lower than that of the wild-type yeast cells (Figure 2A). This result might be due to the lack of suitable substrate in the transgenic yeast cells. Consequently, we hypothesized that the tree peony seed oil was rich in fatty acids such as OA (18:1), LA (18:2), and ALA (18:3). It may be a suitable substrate for *PrDGAT3*. We tested this hypothesis using the principle of yeast lipotoxicity. Subsequently, these exogenous fatty acids were separately added to the culture medium to grow yeast cells.

We observed the growth curve of the yeast for 48 h (Figure 3). We detected no significant difference in the OD between the control and the H1246 yeast expressing *PrDGAT3*, when cultured in medium without any exogenous UFAs. As expected, after the exogenous addition of UFAs, the H1246 yeast cells containing an empty vector grew slowly at the early stage, and the growth tended to stop after 22 h. However, the growth rate of H1246 transgenic yeast cells expressing *PrDGAT3* increased sharply after 35 h of culture, especially when the exogenous LA and ALA were added, suggesting that *PrDGAT3* can overcome the lipotoxicity of yeast by catalyzing the biosynthesis of TAG with exogenous UFAs and may be selective for substrates containing LA and ALA, especially ALA.

### 2.5. Transient Expression of PrDGAT3 Enhances oil and ALA Content in Nicotiana Benthamiana Leaves

To further examine the role of *PrDGAT3* in lipid synthesis and the selective collection of UFAs, we chose *N. benthamiana* as a transient overexpression system. Previously, it was reported that the Agrobacterium-mediated transient transformation had a higher time efficiency than the stable transformation [31]. In addition, the *N. benthamiana* leaf tissue was suitable for the transient expression of the genes involved in fatty acid synthesis and TAG accumulation [32]. Here, we co-transformed the coding sequences of *PrDGAT3* and viral protein *P19*, an inhibitor of ectopic gene silencing, into *N. benthamiana* leaves by Agrobacterium-mediated osmosis. The expression of *PrDGAT3* in *N. benthamiana* was detected by semi-quantitative RT-PCR with the *NbActin* gene of *N. benthamiana* as an internal reference. As shown in Figure 4A, *PrDGAT3* was expressed in *N. benthamiana* leaves at 6 days after Agrobacterium infection. Overexpression of *PrDGAT3* significantly increased the number of lipid droplets (LDs) relative to the control leaves transformed with P19 alone (Figure 4B,C). It is worth noting that there was a considerable increase in the numbers of small-, medium-, and large-sized LDs of the *N. benthamiana* leaves expressing *PrDGAT3*, compared with the P19 control, by 1.8-, 2.1-, and 2.2-fold, respectively (Figure 4D).

Then, we lyophilized the leaves and extracted the lipids for methyl esterification. GC analysis of the fatty acids (Figure 4E) showed that the expression of the *PrDGAT3* gene in the tobacco leaf tissue significantly promoted the total oil content of the leaves, which was about twice that of the control leaves transformed with P19 alone. The expression of *PrDGAT3* also resulted in significant changes in the fatty acid composition (Figure 4F). The polyunsaturated fatty acids LA and ALA were significantly increased, with ALA increased by 10% when compared with the control leaves. These results suggested that *PrDGAT3* most likely prefers UFAs as substrates for TAG synthesis.

In conclusion, our transient leaf expression assays showed that heterologous *PrDGAT3* expression in tobacco can significantly increase oil content in leaves and selectively transfer endogenous UFAs to TAG.

### 2.6. Reduced Expression of PrDGAT3 in Paeonia rockii Represses the Oil Accumulation and Affects Fatty Acids Composition

To further verify the function of *PrDGAT3* in oil accumulation, we infected the leaves of *Paeonia rockii* seedlings and inhibited the endogenous expression of *PrDGAT3*, utilizing the virus-induced gene silencing (VIGS) technique. One week after infection, we observed many green fluorescence spots within the leaves inoculated with TRV2-*GFP* and TRV2-*PrDGAT3-GFP* but not in the MOCK leaves under the excitation by blue light (Figure 5A). Furthermore, the expression of TRV1 and TRV2 RNA was identified in the leaves of the inoculated line through RT-PCR (Figure 5C). In addition, the qRT PCR exhibited that the transcription level of *PrDGAT3* in the leaves inoculated with TRV2-*PrDGAT3-GFP* was significantly lower than that of the MOCK and TRV2-*GFP* lines (Figure 5B). Meanwhile, we detected remarkably decreased expressions of fatty acid biosynthesis genes, including MOD1, *FAD2*, *LPAAT1*, *NFYC*, *DGAT1*, *DGAT2*, and *PDAT2*, in the *PrDGAT3*-silenced leaves (Appendix A). It is worth noting that the expressions of *DGAT1*, *PDAT2*, *FAD2*, and *MOD1* significantly decreased with the reduction in the endogenous *PrDGAT3* transcription level (Figure 5B). These results revealed that *PrDGAT3* may interact with other fatty acid biosynthetic genes in the process of fatty acid and oil accumulation.

In addition, GC analysis was performed on the *PrDGAT3*-silenced leaves. Compared with the TRV2-*GFP* control and the blank control, the total oil content in the *PrDGAT3*-silenced leaves decreased significantly to 15.60 mg/g compared with 31.79 mg/g and 29.52 mg/g (Figure 5D). Notably, compared with the other two control groups, the proportion of the UFAs LA and ALA in the *PrDGAT3*-silenced leaves was significantly reduced. The proportion of LA was 16.16% after *PrDGAT3* silencing, compared with 21.11% and 21.18% in the TRV2-*GFP* control and MOCK, respectively. The proportion of ALA was 48.07% after *PrDGAT3* silencing, compared with 52.39% and 53.42% in the TRV2-*GFP* control and MOCK, respectively (Figure 5E). These results further demonstrated that the expression level of *PrDGAT3* can affect the lipid content and fatty acid composition.

### 2.7. Stable Expression of PrDGAT3 Increases Oil, Especially LA and ALA Accumulation, in Transgentic Arabidopsis Rosette Leaves

Transgenetic Arabidopsis lines expressing pCAMBIA1300-*PrDGAT3* were identified and selected for phenotype analysis (Appendix A). Rosette leaves of three independent lines were collected from OE-4, -8, and -10, with relatively high *PrDGAT3* expression (Figure 6A).

The results showed that the overexpression of *PrDGAT3* displayed significantly increased total fatty acid content and different fatty acid composition. Compared with the wild-type *Arabidopsis*, the total fatty acid content in the rosettes of the OE-4, -8 and -10 transgenic lines increased to 26.6 mg/g, 21.1 mg/g, and 30.8 mg/g, respectively (Figure 6B). In addition, the contents of LA in the OE-4, OE-8, and OE-10 transgenic lines were 20.77%, 20.23%, and 22.82%, respectively, which is significantly higher than the 13.75% in the wild type. Similarly, the contents of ALA were 43.44%, 40.29%, and 48.44% in the OE-4, -8, and -10 transgenic lines, respectively, compared with that of 31.0% in the wild type (Figure 6C). Interestingly, the OE seedlings did not show obvious phenotypic or developmental differences compared with the wild type (Appendix A), suggesting that *PrDGAT3* may play a minor role in other physiological processes. Collectively, these results demonstrated that *PrDGAT3* was involved in the regulation of lipid biosynthesis and fatty acid composition.

To further explore the effect of the overexpression of *PrDGAT3* on lipid synthesis in *Arabidopsis Thaliana*, we examined the expression of various fatty acid and TAG biosynthesis genes in rosette leaves with qRT-PCR (Appendix A). Firstly, we analyzed the expression profile of seven key TAG synthesis genes, including *AtDGAT1*, *AtDGAT2*, *AtDGAT3*, *AtGPAT6*, *AtPDAT1*, *AtPDAT2*, and *AtOLEOSIN* (*AtOLEO3*) (Figure 6D). Next, we analyzed the expression levels of *AtLEC1* and *AtLEC2*, two positive regulators in the process of lipid synthesis. We detected about 7- to 13-fold higher transcription levels of LEC1 and 1.5- to 5-fold higher transcription levels of LEC2 in the OE-4, OE-8, and OE-10 transgenic lines compared with that of the wild type. Finally, we quantified the expression levels of four key fatty acid biosynthetic genes *AtSUS2, AtKASI, AtMOD, and AtFAD2*. Compared with the wild type, the expression levels of most genes were slightly different from those of the wild type, but interestingly, the expression level of *AtSUS2* in the rosette leaves of the transgenic lines was significantly increased. In addition, the expression level of *AtDGAT2* in the rosette leaves of the transgenic lines was significantly decreased. However, the expressions of *AtDGAT1* and *AtPDAT2* were increased significantly. More notably, the expression of *AtOLEOSIN* (*AtOLEO3*) increased 10–24 times in the rosette leaves of the transgenic lines. All together, these findings corroborated that *PrDGAT3* may potentially interact with other genes in lipid synthesis and metabolism during plant development.

## 3. Discussion

Oil content and fatty acid composition are the most important agronomic traits of oil crops. The transcriptional regulation of biosynthetic genes is the major factor affecting the TAG biosynthesis. In previous studies, more and more key genes related to lipid biosynthesis and accumulation have been revealed. *DGAT1* was highly expressed in the developing embryos of oil crops, and its expression level was related to oil accumulation during seed development [12,33]. The inactivation of *DGAT1* resulted in reduced seed oil content in *Arabidopsis* mutant AS11 [34]. The co-transformation of *VfFAD2* and *VfDGAT2* genes in the oilseed of *Vernicia fordii* resulted in a 17.10% increase in Eicosatenoic Acid (C20:3) in *Arabidopsis* [35]. Past studies have shown that DGATs are the ultimate rate-limiting enzymes for TAG biosynthesis in eukaryotes [36,37]. Compared with the commonly studied *DGAT1* and *DGAT2* gene family members, *DGAT3* gene has only been identified in a few plant species, and its function is still unknown [18,27,28,29].

In this study, we identified and cloned *PrDGAT3* from *Paeonia rockii*. One or two *DGAT3* genes have been reported in the genomes of various plants [26], ranging from lower prokaryotes such as *Chlamydomonas reinhardtii* and *Phaeodactylum tricornutum* to important cash crops such as soybean, corn, and camelina. Two isoforms of the DGAT3 family were identified from the genome of oil palm (*Elaeis guineensis*) [38], and three isoforms were identified from peanut [17]. The structural analysis of *DGAT3* proteins showed that most plant DGAT3 proteins were located in the cytoplasm and were water-soluble proteins. In agreement with this finding, our study revealed that *PrDGAT3* of *P. rockii* was a water-soluble protein without a transmembrane structure. Phylogenetic analysis (Figure 1C) showed that the PrDGAT3 protein of *P. rockii* had the highest homology with the DGAT3 protein of the genus Vitis, suggesting that the two species may have shared a common ancestor during evolution. Similarly to the other DGAT3s, PrDGAT3 also contained a TRX-like Fd domain, which was a conserved thioredoxin-like ferredoxin domain located in the C-terminal region of the DGAT3s, and it contained four conserved cysteines involved in [2Fe-2S] cluster binding. At the same time, this domain indicated that DGAT3 was a metalloprotein, and its function may be related to iron utilization. Consistently with this, the expression of *AtDGAT3* was up-regulated in the leaves and roots of *Arabidopsis* under iron stress [39], while the over-accumulation of iron led to the down-regulation of the *AtDGAT3* expression in flowers [40]. On the other hand, it is speculated that *DGAT3* had a dual activity in which electron transfer was coupled to the acylation reaction [41]. Furthermore, the DGAT3 conserved motif was mostly located in the N-terminal region, indicating that it was the most conserved region. The presence of multiple functional motifs in the PrDGAT3 protein may indicate that their function was influenced or regulated by other factors. Therefore, further experiments are needed to elucidate the exact role of this domain in the DGAT3 protein.

Interestingly, the transiently expressed PrDGAT3 protein was localized in the chloroplasts of tobacco leaf cells (Figure 1A), which was consistent with the prediction of the chloroplast transport peptide in PrDGAT3 protein. Notably, the DGAT3 activity has also been detected in the cytosolic and plastid fractions [18], predicting an N-terminal chloroplast transport peptide in AtDGAT3 [27]. In higher plants, TAG accumulates as lipid droplets in the cytoplasm and in plastids [42,43], which include chloroplasts, chromatids, and alba bodies [44,45,46]. An in vitro assay has shown that even a small increase in ALA can cause severe damage to membrane structure and function within the chloroplasts; so, chloroplasts are highly sensitive to free fatty acids [47,48,49]. One of the primary functions of LDs is to buffer the levels of cytotoxic lipids and release stored neutral lipids in response to membrane homeostasis and cell growth requirements [50]. Therefore, in contrast to the TAG in the cytoplasm, the main function of the TAG in chloroplasts is to reduce the damage caused by lipotoxicity, thereby reducing the direct damage caused by lipids to the organism. In addition, DGAT1, a key rate-limiting enzyme involved in TAG synthesis in the endoplasmic reticulum, has been shown to be responsible for the chloroplast envelope [51,52], and the activity of *Arabidopsis PDAT1* has also been shown to be associated with the microsomal membrane [11]. However, to date, the relationship between the cytoplasm and the chloroplast TAG metabolism is still unclear. It is crucial to explore the metabolism of TAG in the chloroplasts and the enzymes that catalyze the final step of the TAG assembly [53]. Collectively, these data showed that DGAT3 subcellular localization is different from that of higher plants; hence, future experimentations are required to illustrate the exact role of *PrDGAT3* in chloroplast TAG synthesis and its association with cytoplasmic TAG metabolism.

Previous studies have shown that DGAT3 has different substrate selection for fatty acids in various plant species. For example, the transient expression of *AtDGAT3* in tobacco leaves resulted in increased TAG accumulation compared with the *AtDGAT1*-expressing control, which significantly enhanced the LA and ALA accumulation [14], and this activity was also verified by the in vitro assay of bacterially expressed *AtDGAT3* [27]. *CsDGAT3-3* in camellia seeds was functionally verified by the *Saccharomyces cerevisiae* system, in the in vitro and tobacco experiments. The results showed that *CsDGAT3-3* had high DGAT activity against UFAs, especially against 20:1n-9. The *RgDGAT3* gene from the *Rhodotorula glutinis* [54] preferentially selected for C18 UFAs in restoring TAG synthesis in the *Saccharomyces cerevisiae* mutant H1246. In our study, the functional complementation experiment in the yeast strain H1246 and the yeast lipotoxicity assay, as well as the heterogeneously transient expression in tobacco leaves and the stable expression in *Arabidopsis Thaliana* and *PrDGAT3* gene silencing, demonstrated that *PrDGAT3* has a high DGAT activity and substrate preference for UFAs, particularly LA and ALA.

Both the transient and the stable overexpression of *PrDGAT3* induced a significant increase in the total oil content of the plant and altered its fatty acid composition (Figure 4A,C and Figure 6B,C). In addition, the silencing of *PrDGAT3* resulted in lower total plant oil content and the composition of several major fatty acids (Figure 5B,C). The expression of genes involved in the regulation of lipid biosynthesis is critical for TAG accumulation. With qRT-PCR, we found that overexpression or silencing of *PrDGAT3* affected the expression of several key genes involved in TAG biosynthesis. For example, *AtOLEO3* is an oil body protein, which has important effects on fatty acid biosynthesis. The expression level of *AtOLEO3* was significantly increased in the *PrDGAT3* overexpression lines. As key rate-limiting enzymes in the last step of TAG biosynthesis, the expression levels of *AtDGAT1* and *AtPDAT2* increased significantly in the *PrDGAT3* overexpressing lines. Similarly, the expression levels of *PrDGAT3* decreased when *PrDGAT3* was silenced, indicating that *DGAT3* may have a potential relationship with *DGAT1* and *PDAT2* in plants. LEC1 and LEC2, are important in regulating seed maturation and lipid accumulation in *Arabidopsis* [55]. More interestingly, we found to our surprise that the expression levels of *LEC1* and *LEC2* increased in the *PrDGAT3* overexpression lines. Further studies are needed to determine whether *DGAT3* is related to *LEC1* and *LEC2*.

## 4. Materials and Methods

### 4.1. Plant Materials and Regents

The *Paeonia rockii* used in the experiment was grown in the tree peony Germplasm Resource nursery of Northwest A&F University in Shaanxi Province, China. The *P. rockii* was managed from seedling planting to seed harvest. The roots, stems, leaves, petals, and seeds of *P. rockii* were collected in liquid nitrogen and then stored at −80 °C. Two-year-old seedlings were used for virus-induced gene silencing (VIGS). *Escherichia coli* DH5α, *Agrobacterium tumefaciens* GV3101, and yeast (*Saccharomyces cerevisiae*) wild-type strain (INVSc1) were purchased from Shanghai Viland Biotechnology Company. The yeast TAG-deficient strain H1246 was generously donated by professors of Hainan University. The yeast expression vector pYES2 was stored in the Oil Tree Peony Engineering Center of Northwest A&F University. The yeast transformation kit and yeast plasmid extraction kit were purchased from Beijing Coolaber Technology Company. The plant expression vectors pCAMBIA1300 and pCAMBIA2300-GFP and the gene silencing vector TRV2-*GFP* were preserved by the Oil Tree Peony Engineering Center of Northwest A&F University. The toxicity test of the yeast fatty acids was carried out by adding a single fatty acid with a concentration of 1 mM into the culture medium. The long-chain fatty acids used included oleic acid (OA, C18:1, 99% purity), linoleic acid (LA, C18:2, 99% purity), and α-linolenic acid (ALA, C18:3, 98% purity), purchased from Beijing Solebo Technology Company.

### 4.2. The Cloning of PrDGAT3 and Sequence Analysis

We found the DGAT3 gene fragment in the previous transcriptome data and cloned the full-length gene by the chromosome step. The full-length CDS of *PrDGAT3* was cloned from *P. rockii* (Appendix A). Based on the protein sequence, we constructed an unrooted phylogenetic tree of the PrDGAT3 and 18 DGAT3 protein sequences from other plant species with MEGA6.0, using the neighbor-joining (NJ) method (bootstrap replicates:1000); the GenBank accession numbers of the sequences are listed in Appendix A. The basic physical and chemical properties of the PrDGAT3 protein were analyzed. The hydrophilicity, molecular weight, and isoelectric point of the PrDGAT3 homologous proteins were predicted using the online analysis website (http://www.expasy.org, accessed on 8 March 2022). The transmembrane (TM) structure of the PrDGAT3 proteins was predicted using the online software TMHMM Server v.2.0 (http://www.cbs.dtu.dk/services/TMHMM/, accessed on 8 March 2022). The amino acid sequence of *PrDGAT3* was submitted to the MEME program to identify conserved protein motifs [56]. The amino acid sequence of *PrDGAT3* was analyzed by the PRALINE program (http://www.ibi.vu.nl/programs/Pralinewww/, accessed on 8 March 2022).

### 4.3. Expression Analysis of PrDGAT3

The qRT-PCR primers were designed by Oligo6.0, and *P. rockii*’s 18S-26S internal transcribed spacer (18S-26S ITS) gene [57] was used as the internal reference gene, as shown in Appendix A. RNA from the seeds and other organs was extracted using the RNAprep Pure Plant kit (Tiangen Beijing, China); subsequently, approximately 1 µg of total RNA was used for reverse transcription, using the PrimeScript^TM^ RT reagent Kit (TaKaRa Dalian, China). Three copies of each sample were analyzed. The relative expression analyses were performed by the 2−ΔΔCT values [58]. The reverse transcription of the *P. rockii* cDNA was used as a template for PCR amplification. The PCR products were purified by the DNA Gel Extraction Kit (Sangon), and then, the amplified PCR fragments were inserted into the pMD19-T vector. After that, positive clones were screened for gene sequencing. The correctly sequenced plasmids were stored in the refrigerator at −20 °C for later usage.

### 4.4. Construction of PrDGAT3 Expression Vectors

The plasmid pCAMBIA1300-*PrDGAT3* was constructed by inserting the CDS of *PrDGAT3* into pCAMBIA1300. The vector pCAMBIA2300-*PrDGAT3-GFP* was generated for subcellular localization by inserting the CDS of *PrDGAT3* without the stop codon into the pCAMBIA2300-*GFP* vector. The gene fragment was amplified and linked to TRV2 to construct TRV2-*PrDGAT3*-*GFP* vectors for VIGS analysis. To construct the yeast overexpression vector, we inserted the CDS of *PrDGAT3* into the yeast expression vector pYES2. The primers used for gene cloning and vector construction in this study were designed by Oligo 6.0 software and are shown in Appendix A.

### 4.5. Subcellular Location

To identify the subcellular location of PrDGAT3, the pCAMBIA2300- *PrDGAT3-GFP* vector was transiently transformed into tobacco leaf cells. The recombinant plasmid pCAMBIA2300-*PrDGAT3-GFP* was first transferred into the receptor cells of *Agrobacterium tumefaciens* GV3101 by the freeze–thaw method [32]. The Agrobacteria were cultured until OD600, reaching 0.4 in the YEP medium containing kanamycin (50 µg/mL) and rifampicin (50 µg/mL), suspended with equal volume buffer (10 mM MES, 200 uM Acetosyringone, and 10 mM MgCl2), and then injected into 4-week-old tobacco leaves through the lower epidermis. After 4–6 days, the injected tobacco leaves were scanned for GFP signals under a confocal laser scanning microscope (LEICA TCS SP8, Germany Heidelberg) with an excitation wavelength of 488 nm and a spontaneous light detection wavelength of 640 nm.

### 4.6. Heterogeous Expression in the S. cerevisiae Mutant H1246 Strain

The recombinant plasmid (pYES2-*PrDGAT3*) was transformed into yeast strain H1246 [59] by the LiAc method. The empty pYES2 vector was transformed into wild-type yeast strain INVSc1 and yeast strain H1246, as the positive control and negative control, respectively. The positive colonies were screened on the lowest medium plate lacking uracil (SC-U), and the positive colonies were screened by PCR (primers are in Appendix A). Firstly, the recombinant yeast cells were grown in 10 mL SC-URA inhibitory medium with 2% (*w/v*) glucose overnight at 200 r/min at 30 °C and then inoculated at a starting point of OD600 of 0.1 in a minimum medium containing 2% (*w/v*) galactose and 1% (*w/v*) raffinose (referred to as the induction medium) to induce gene expression (200 r/min; 30 °C for 36 h). The cells were collected by centrifugation (5000× *g*, 8 min) and washed with double-distilled water three times. The cells used for the subsequent lipid analysis were dried in a freeze dryer and stored at room temperature. 

### 4.7. Yeast Fatty Acid Toxicity Test for acyl-CoA Substrate Specificity Assay 

A yeast fatty acid toxicity test was employed to determine the amount of free fatty acids added to a culture medium that can produce lipid toxicity for TAG-deficient yeast. Nevertheless, if the transgenic yeast can produce acyltransferase activity that can selectively convert free fatty acids and restore TAG accumulation, it can overcome lipid toxicity and grow normally [60]. We evaluated the effects of exogenous fatty acids on total TAG content and fatty acid composition in yeast expressing *PrDGAT3*; at the same time, we identified the substrate preference of *PrDGAT3*. To scatter the fatty acids into the medium, a solution of 1 mM fatty acid solution was first dissolved in 0.5 M ethanol and then diluted in a warm medium containing 0.06% (*v/v*) tyloxapol (Sigma) before mixing with the induction medium. Approximately 30 mg of dry yeast was used for culturing in an induction medium supplemented with different fatty acids (1 mM) OA, LA, and ALA. The yeast cultured in an induction medium without exogenous fatty acid was used as the control group. Then, the yeast was placed in a 28 °C shaking box at 200 r/min for 48 h. During this time, the cell density was measured by UV spectrophotometer at different time points to make the yeast growth curve (A600 nm).

### 4.8. Lipid Extraction, Fatty Acid Analysis, and Visualization of Lipid Droplets

Lyophilized yeast was ground into powder, and the total lipid was extracted using the n-hexane: isopropanol (3:2) method [61]. A solvent mixture of n-hexane: diethyl ether: acetic acid (80:20:2, *v/v/v*) was used to separate the labels from the total lipids by thin layer chromatography (TLC). The separated TAG bands were scraped off the silica gel plate and eluted with 2 mL n-hexane for 2 h. The supernatant passed through a 0.22 μm organic filter and was placed in a new centrifuge tube. In accordance with Zhang [62], the fatty acid composition analysis of fatty acid methyl ester was carried out with C17:0 as an internal standard. Three copies of each sample were analyzed [63]. Fatty acid methyl ester (FAME) was analyzed by the Agilent 8890 gas chromatography system (Agilent, Santa Clara, CA, USA), equipped with an HP-Innowax (60 m × 0.25 mm, 0.25 µm) elastic quartz capillary column with nitrogen as a carrier gas at a 250 °C inlet temperature and a 130 °C chamber start-up temperature.

The Nile red fluorescence assay was applied to observe intracellular lipid droplets using the confocal laser scanning microscope (LEICA TCS SP8, Germany Heidelberg). A 1 mL yeast cell culture was obtained by centrifugation (5000× *g* for 10 min) for 36 h of growth and was re-suspended with double-distilled water to OD600 = 0.5. Then, 200 μL yeast cells was stained with 10 μL Nile red (5 mg/mL methanol), 50 μL dimethyl sulfoxide, and 750 μL double-distilled water in the dark for 15 s [29]. Nile red was excited at 488 nm with emission wavelengths of 560–620 nm and then observed by confocal microscopy.

### 4.9. Transient Overexpression in Nicotiana Benthamiana Leaves

The recombinant plasmid pCAMBIA1300-*PrDGAT3* was transferred into the competent cells of *Agrobacterium tumefaciens* GV3101 by the freeze–thaw method. The positive colonies were selected by COLONY PCR after being cultured for 48 h on a solid YEP medium containing kanamycin (50 µg/mL) and rifampicin (50 µg/mL) under dark conditions at 28 °C. The positive colonies were then sub-cultured in liquid medium containing the same antibiotic of YEP and continued to be shaken overnight at 28 °C. The Agrobacteria cells were then collected by centrifugation at 8000 r/min for 10 min and suspended in osmotic buffer (300 µM acetosyringone, 10 mM MES, 10 mM MgCl2, pH 5.7) until the final OD600 was 0.4–0.6. The culture was kept in the dark at 28 °C for 3 h. In order to improve the expression of *PrDGAT3* in the tobacco leaves, agrobacterium suspensions containing P19 silencing inhibitors and pCAMBIA1300-*PrDGAT3* suspensions were mixed in a 1:1 ratio and injected into the reverse side of 4-week-old tobacco leaves with sterile syringes. The leaves injected with the *Agrobacterium tumefaciens* P19 were used as the control. The tobacco plants injected with the Agrobacteria solution were placed in the culture chamber. On the fourth day, some tobacco leaves were dyed with Nile red for the lipid droplet visualization experiment. The lipid droplets were stained with 2µg/mL Nile red (raw solution) in 0.01 mM phosphate buffer (pH 7.2). A confocal laser scanning microscope (LEICA TCS SP8, Germany) was used to obtain the confocal images according to the method described above. On the fifth day, approximately 0.1 g of tobacco leaves per replicate was weighed out after stabilizing the leaf moisture content in a freeze drier for 72 h; three biological replicates were prepared for each sample. Then, the leaves were placed in a 10 mL glass test tube and a 4 mL 2.5% methanol sulfate solution was added first; then, they were put into a water bath at 80 °C for 2 h in the dark. After dropping to room temperature, 1 mL 0.9% sodium chloride solution and 1 mL n-hexane were added sequentially. After shaking and full extraction and 250× *g* centrifuge for 5 min, the supernatant was isolated with 0.22 µm organic filter membrane placed into the sample bottle for GC (gas chromatography) detection. The gas chromatographic conditions were the same as those described above.

### 4.10. Virus-Induced Gene Silencing of PrDGAT3

The seedlings of *Paeonia rockii* were dyed according to the method of Xie [64]. TRV2-*PrDGAT3*-*GFP*, TRV2-*GFP*, and TRV1 were transformed into *Agrobacterium* GV3101, respectively. The transformed GV3101 were grown in YEP medium containing kanamycin (50 µg/mL), rifampicin (50 µg/mL), and gentamicin (50 µg/mL) at 28 °C with constant shaking overnight. After the supernatant was dumped, the cells were treated with Acetosyringone containing 10 mM morpholinoethulfonic Acid (MES) and 100 μM Acetosyringone (AS) and 10 mM MgCl2 immersion buffer to re-suspend the Agrobacteria and adjust the OD600 to 1.2~1.5. The Agrobacteria solution containing TRV1 was mixed with the Agrobacteria solution containing TRV2-*GFP* and TRV2-*PrDGAT3-GFP* in equal volumes, and then, the mixed bacterial solution was placed at 28 °C, 90 r/min and sheltered from light for 3 h. The leaves of the *P. rockii* seedlings infected with TRV2-*GFP* were used as a control. After 6 days of infection, the leaves were used for *GFP* expression analysis. After 14 days of infection, the leaf samples of the *GFP*-expressed plants were collected for subsequent fatty acid content determination and gene expression evaluation with real-time quantitative PCR. The specific methods were described above.

### 4.11. Overexpression of PrDGAT3 in Arabidopsis Thaliana

The recombinant plasmid pCAMBIA1300-*PrDGAT3* was transformed into *Agrobacterium tumefaciens* strain GV3101. Subsequently, the Agrobacterium strains were used to transform the wild-type *Arabidopsis* (Columbia) by the floral dipping method [65]. T0 seeds were germinated on 1/2 MS medium containing hygromycin (20 mg/l) to screen the positive transformed *Arabidopsis* lines. The seedlings with normal root growth were cultured in the substrate for two to three weeks before the collection of the young leaves. Semi-quantitative RT-PCR was used to identify the seedlings transformed with *PrDGAT3* (Appendix A). These transgenic lines were grown into a T2 generation to obtain homozygous lines. Mature Arabidopsis leaves of the T2 transgenic lines were collected and randomly selected for different lines; the fatty acids were extracted for analysis. The specific methods were described above.

## 5. Conclusions

In this study, we systematically dissected the phylogeny, subcellular localization, and functional roles of PrDGAT3 in *Paeonia rockii*, a promising woody oil crop. *PrDGAT3* is mainly expressed in the vegetative organs, such as stems and leaves. With protein motif analysis, we found that PrDGAT3 protein shares a conserved domain with the same family members of other species. By integrating functional assays in yeast mutant H1246 with and without exogenous fatty acid feeding, as well as *PrDGAT3* gene silencing, the transient expression in tobacco leaves and the stable expression in *Arabidopsis Thaliana*, we detected that *PrDGAT3* has a high DGAT activity and a substrate selection for UFAs, particularly linoleic acid (LA, C18:2) and α-linolenic acid (ALA, C18:3). Our findings provide new knowledge to advance our understanding of the regulation of TAG biosynthesis and metabolism in plants, and in particular, they provide a new insight into the genetic improvement of oil yield and quality through biotechnology.

## Figures and Tables

**Figure 1 ijms-23-14390-f001:**
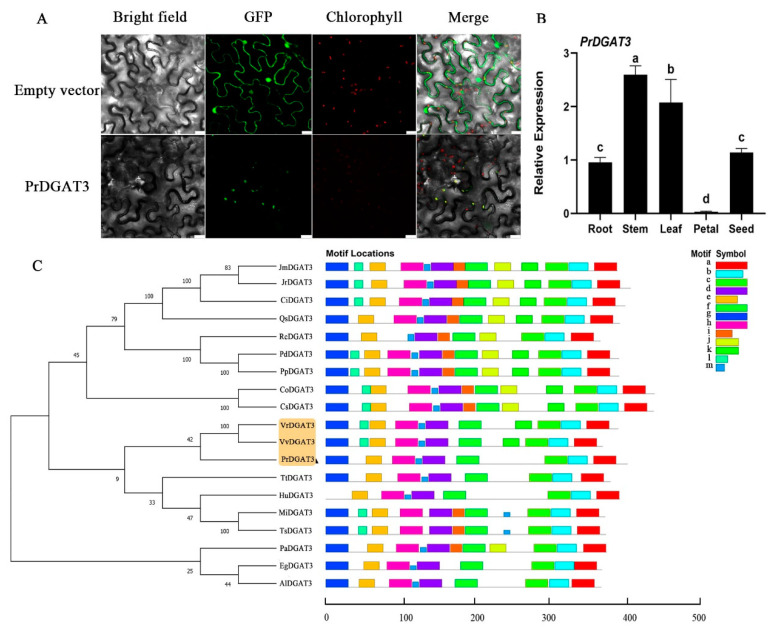
Expression profile and molecular characterization of PrDGAT3. (**A**) Subcellular localization of the PrDGAT3 protein fused with GFP (pCAMBIA2300- *PrDGAT3-GFP*) in tobacco leaf cells. Bars = 20 µm. (**B**) *PrDGAT3* expression in various tissues of *P. rockii.* Values are mean ± SD (*n* = 3)**.** (**C**) Phylogenetic tree and conserved motifs of the PrDGAT3. Different letters indicate significant difference at *p* < 0.05, as confirmed by one-way ANOVA with Tukey’s post hoc test.

**Figure 2 ijms-23-14390-f002:**
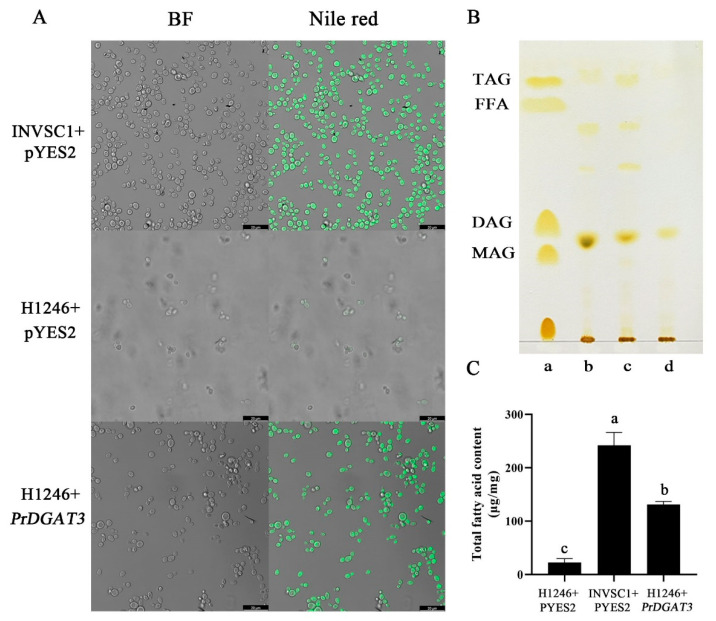
Functional verification of *PrDGAT3* in yeast mutant H1246. (**A**) Oil body formation in yeast cells stained with Nile red and examined by confocal laser scanning microscope. Bars = 20 µm. (**B**) Evaluation of TAG biosynthesis in the transgenic yeast cells by thin-layer chromatography (TLC). Lane a, standards of acyl-lipid classes: TAG (triacylglycerol), FFA (free fatty acids), DAG (diacylglycerols), and MAG (monoacylglycerol). Lane b, neutral lipids from yeast mutant H1246 cells expressing *PrDGAT3*. Neutral lipids from yeast mutant H1246 cells harboring the empty vector (pYES2) (negative controls). Lane c, neutral lipids from the wild-type yeast cells (positive controls). Lane d, neutral lipids from yeast mutant H1246 cells harboring the empty vector (pYES2) (negative controls). (**C**) Total fatty acid content (% dry weight). Values are mean ± SD (*n* = 3). Different letters indicate significant difference at *p* < 0.05, as confirmed by one-way ANOVA with Tukey’s post hoc test.

**Figure 3 ijms-23-14390-f003:**
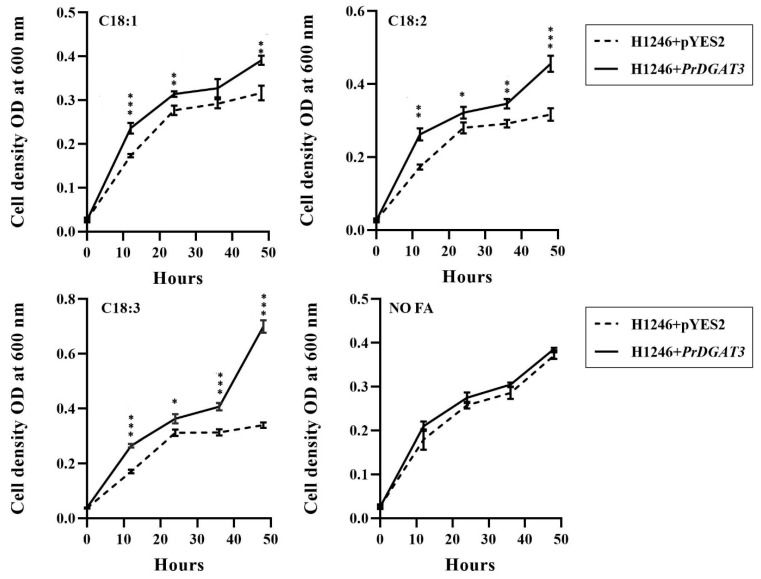
Quantification of the substrate specificity of PrDGAT3 for different acyl-CoAs by yeast lipid toxicity test. OD values of yeast cells under different conditions over time. Values are mean ± SD (*n* = 3). * *p* < 0.05, ** *p* < 0.01, *** *p* < 0.001.

**Figure 4 ijms-23-14390-f004:**
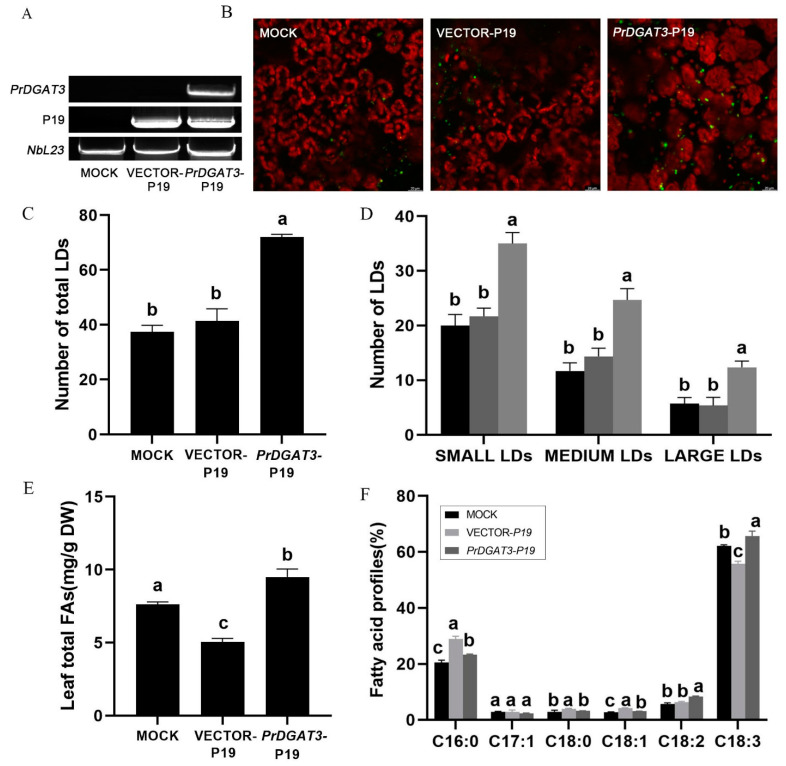
Functional verification of *PrDGAT3* in transformed *N. benthamiana* leaf tissue. (**A**) RT-PCR analysis of *PrDGAT3* expressed in tobacco leaf tissue. (**B**) Representative confocal images of LDs in *N. benthamiana* leaf tissue. Green color shows LDs and red color shows chloroplasts. Images are shown as 232.92 × 232.92 µm^2^ field of the leaf tissue. Bars = 20 µm. (**C**) Number of total LDs per image area in mock, *P19*-transformed, and *PrDGAT3*-transformed *N. benthamiana* leaf tissue. (**D**) Number of LDs in different size categories per image area. Small LDs: Nile red-stained lipid area < 3 µm^2^; medium LDs: 3–6 µm^2^; large LDs: 6–10 µm^2^. (**E**) Total oil content in tobacco leaves (% dry weight). (**F**) Major fatty acid profiles in TAG. Values are mean ± SD (*n* = 3). Different letters indicate significant difference at *p* < 0.05, as confirmed by one-way ANOVA with Tukey’s post hoc test.

**Figure 5 ijms-23-14390-f005:**
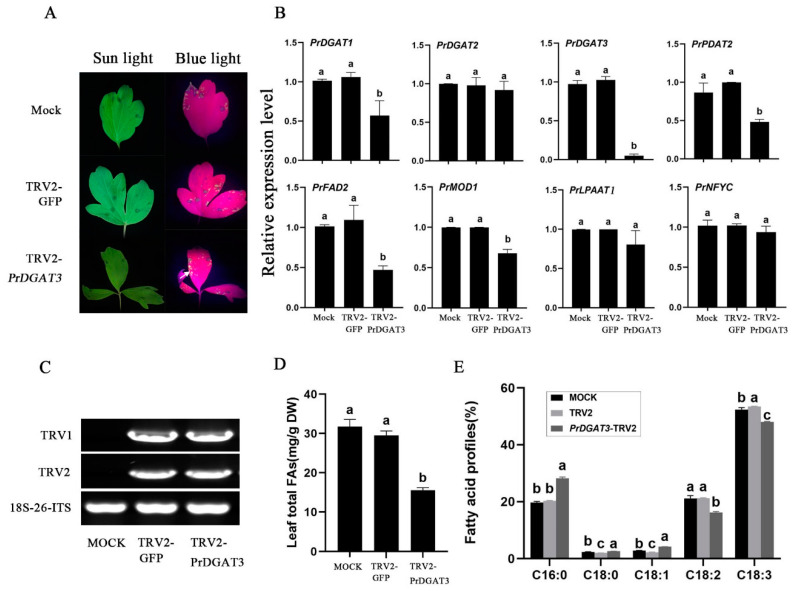
Reduced expression of *PrDGAT3* in *Paeonia rockii* reduces the oil content and affects fatty acid composition as well as relative gene expression. (**A**) Image of *P. rockii* leaves infected with TRV2-*GFP* or TRV2-*PrDGAT3-GFP* at 6 days post-infiltration under blue light. (**B**) Comparison of the expression levels of fatty acid and TAG biosynthetic genes among the Mock, TRV2-*GFP* control, and *PrDGAT3*-silenced plants. (**C**) RT-PCR analysis of TRV1 and TRV2 accumulation levels in *P. rockii* leaf tissue at 2 weeks post-infiltration. (**D**) Total oil content in *P. rockii* leaves among the Mock, TRV2-*GFP* control, and *PrDGAT3*-silenced plants (% dry weight). (**E**) Major fatty acid profiles in TAG among the Mock, TRV2-*GFP* control, and *PrDGAT3*-silenced plants. Values are mean ± SD (*n* = 3). Different letters indicate significant difference at *p* < 0.05, as confirmed by one-way ANOVA with Tukey’s post hoc test.

**Figure 6 ijms-23-14390-f006:**
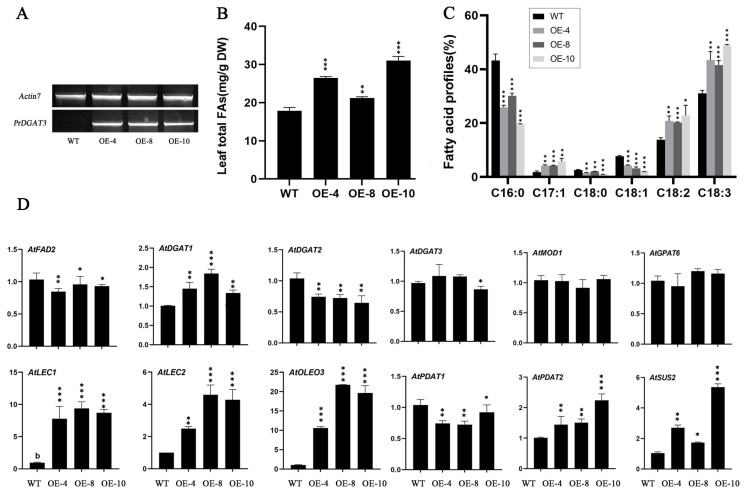
Functional verification of PrDGAT3 in transformed *Arabidopsis*. (**A**) RT-PCR analysis of *PrDGAT3* expressed in transgenic *Arabidopsis* rosette leaves. (**B**) Total oil content in *Arabidopsis* rosette leaves between the wild-type (Col-0) and the *PrDGAT3* transgenic plants (% dry weight). (**C**) Major fatty acid profiles in TAG between the wild-type (Col-0) and *PrDGAT3* transgenic plants. (**D**) Comparison of the expression levels of fatty acid and TAG biosynthetic genes between the wild-type (Col-0) and *PrDGAT3* transgenic plants. Values are mean ± SD (*n* = 3). * *p* < 0.05, ** *p* < 0.01, *** *p* < 0.001.

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
