# Peer review of "Diacylglycerol Acyltransferase 3(DGAT3) Is Responsible for the Biosynthesis of Unsaturated Fatty Acids in Vegetative Organs of Paeonia rockii"

_ijms, 2022, doi:10.3390/ijms232214390_

Round 1
Reviewer 1 Report
The authors have carried out an array of experiment to evaluate the role the enzyme Diacylglycerol acyltransferase 3(DGAT3) plays in the Paeonia rockii. As summarized by the authors, the role of t this enzyme has been studied in the context of various cell plants, and its importance in the synthesis of unsaturated fatty has been established. In this work, the authors extended this work by using another species of nutritional importance, Paeonia rockii.
To the best of my knowledge, this methodology is sound, the results are convincing, and overall the data have been well discussed. So I do not major comments on this work. I only have few comments that can improve the readability of the paper.
General comments:
1. The paper is full with abbreviations, so it can become difficult to read. So I suggest that the number of abbreviations be reduced to only those that are repeated several times in the text. If and abbreviation is not repeated many times then it can be removed, and the name be written completely. For instance, G3P was not repeated in the text, (GPAT) was repeated only 2 times. So the authors could see if such abbreviations could be removed.
Specific comments:
2. Line 23: LA and ALA to be explained here.
3. Line 43-53: I would suggest that a biochemical pathway diagram of the synthesis of triacid glycerol (TAG) be made here, so that to make it clearer for the reader, specially those who are not expert in this topic.
4. Line 50: DGAT is used widely in the text, however, this is abbreviation is not explained at all, so it is difficult to the follow the rest of the manuscript!!
Author Response
Thank you very much for your valuable advice! Please find the attachment for details.

Author Response

(The authors gave the same response as above.)

Round 2
Reviewer 2 Report
Well done.